# The Use of an Antioxidant Enables Accurate Evaluation of the Interaction of Curcumin on Organic Anion-Transporting Polypeptides 4C1 by Preventing Auto-Oxidation

**DOI:** 10.3390/ijms25020991

**Published:** 2024-01-12

**Authors:** Toshihiro Sato, Ayaka Yagi, Minami Yamauchi, Masaki Kumondai, Yu Sato, Masafumi Kikuchi, Masamitsu Maekawa, Hiroaki Yamaguchi, Takaaki Abe, Nariyasu Mano

**Affiliations:** 1Department of Pharmaceutical Sciences, Tohoku University Hospital, Sendai 980-8574, Japan; masaki.kumondai.d5@tohoku.ac.jp (M.K.); yu.sato.e7@tohoku.ac.jp (Y.S.); masafumi.kikuchi.b2@tohoku.ac.jp (M.K.); m-maekawa@tohoku.ac.jp (M.M.); mano@hosp.tohoku.ac.jp (N.M.); 2Graduate School of Pharmaceutical Sciences, Tohoku University, Sendai 980-8578, Japan; 3Department of Pharmacy, Yamagata University Hospital, Yamagata 990-9585, Japan; hiroaki.yamaguchi@med.id.yamagata-u.ac.jp; 4Graduate School of Medical Science, Yamagata University, Yamagata 990-9585, Japan; 5Division of Nephrology, Endocrinology, and Vascular Medicine, Graduate School of Medicine, Tohoku University, Sendai 980-8574, Japan; takaabe@med.tohoku.ac.jp; 6Division of Medical Science, Graduate School of Biomedical Engineering, Tohoku University, Sendai 980-8579, Japan; 7Department of Clinical Biology and Hormonal Regulation, Graduate School of Medicine, Tohoku University, Sendai 980-8575, Japan

**Keywords:** OATP4C1, renal transporter, food–drug interaction, flavonoids, curcumin, kaempferol, ascorbic acid

## Abstract

Flavonoids have garnered attention because of their beneficial bioactivities. However, some flavonoids reportedly interact with drugs via transporters and may induce adverse drug reactions. This study investigated the effects of food ingredients on organic anion-transporting polypeptide (OATP) 4C1, which handles uremic toxins and some drugs, to understand the safety profile of food ingredients in renal drug excretion. Twenty-eight food ingredients, including flavonoids, were screened. We used ascorbic acid (AA) to prevent curcumin oxidative degradation in our method. Twelve compounds, including apigenin, daidzein, fisetin, genistein, isorhamnetin, kaempferol, luteolin, morin, quercetin, curcumin, resveratrol, and ellagic acid, altered OATP4C1-mediated transport. Kaempferol and curcumin strongly inhibited OATP4C1, and the *K*_i_ values of kaempferol (AA(−)), curcumin (AA(−)), and curcumin (AA(+)) were 25.1, 52.2, and 23.5 µM, respectively. The kinetic analysis revealed that these compounds affected OATP4C1 transport in a competitive manner. Antioxidant supplementation was determined to benefit transporter interaction studies investigating the effects of curcumin because the concentration-dependent curve evidently shifted in the presence of AA. In this study, we elucidated the food–drug interaction via OATP4C1 and indicated the utility of antioxidant usage. Our findings will provide essential information regarding food–drug interactions for both clinical practice and the commercial development of supplements.

## 1. Introduction

In recent years, polyphenols have received considerable attention owing to their antioxidant, antiviral, anti-inflammatory, and anticarcinogenic activities [1,2,3,4,5]. Polyphenols have been reported to prevent major chronic diseases, including cardiovascular diseases, cancer, and diabetes [6,7,8]. However, some flavonoids, a group of natural polyphenolic substances abundant in vegetables, fruits, grains, and tea, interact with drugs via transporters and metabolic enzymes [9,10,11,12,13,14] (Appendix A).

Drug interactions via transporters and metabolic enzymes are considered the main cause of adverse drug reactions [15,16]; thus, drug–drug and drug–endogenous metabolite interaction studies have been widely conducted [17,18,19,20,21]. Quercetin, a flavonoid, has been reported to increase the plasma concentration of pravastatin by inhibiting organic anion-transporting polypeptide (OATP) 1B1 [22]. Consequently, drug transporters, such as OATP, have been investigated, and information on food–drug interactions has increased (Appendix A). For example, the flavonoid ellagic acid inhibits organic anion transporter (OAT) 1 expressed in renal proximal tubular cells [23]. 

OATPs are Na^+^-independent transporters expressed in various tissues, such as the liver, kidney, and small intestine, and contribute to the cellular membrane transport of endogenous substances, such as bile acids and various drugs [17,24,25,26,27]. OATP4C1 is the only member of the OATP family expressed in the kidney, is localized on the basolateral side of proximal tubule cells, and has been reported to contribute to the renal excretion of uremic toxins and some drugs [28,29]. Recently, OATP4C1 has been reported to interact with human immunodeficiency virus protease inhibitors such as ritonavir and saquinavir [18], and remdesivir, an antiviral drug used for the treatment of COVID-19 [19]. 

Although some flavonoids have been reported to inhibit OATP1B1, OATP1B3, and OATP2B1 [30,31], the inhibition of OATP4C1 has not been studied extensively. The aim of this study was to investigate the food–drug interactions via OATP4C1 to understand the safety profile of food ingredients, including flavonoids, in renal drug excretion.

## 2. Results

### 2.1. Screening of the Effect of Food Ingredients on OATP4C1-Mediated Transport

To elucidate potential food–drug interactions via OATP4C1, we screened 28 food ingredients (Figure 1 and Figure 2) that are commonly used in drug interaction studies (Appendix A). Some of these food ingredients are also reported to be the substrates or inhibitors of organic ion transporters and ABC transporters. Inhibitor concentrations were set much higher (100 µM, except for theaflavin [50 µM] due to the solubility upper limit) than those expected in human blood to avoid overlooking the interaction. OATP4C1-mediated transport was strongly inhibited (by > 50%) by six compounds: apigenin, kaempferol, luteolin, morin, quercetin, and curcumin (Figure 3). These compounds exhibited 66%, 75%, 52%, 57%, 50%, and 72% inhibition, respectively. Six other compounds, namely daidzein, fisetin, genistein, isorhamnetin, resveratrol, and ellagic acid, also altered OATP4C1-mediated transport; however, the alteration by these compounds was moderate (> 10% and < 50%) (Figure 3). These compounds exhibited 40%, 24%, 42%, 18%, and 46% inhibition and 13% elevation, respectively. No significant alterations in OATP4C1-mediated transport were observed for the other 16 food ingredients screened (Figure 3).

### 2.2. Using an Antioxidant to Evaluate the Inhibitory Effects of Food Ingredients

As curcumin is unstable in phosphate buffer at physiological pH [32], ascorbic acid (AA) was used to prevent curcumin degradation during the transport study. We confirmed that AA did not affect OATP4C1-mediated transport (Figure 4A). Next, we evaluated the inhibitory effect of curcumin in the absence or presence of 10, 100, and 1000 µM AA (Figure 4B). A significant increase in curcumin inhibition, from 33% to 63–67%, was observed (Figure 4B). The addition of AA did not change the pH of the reaction solution (control: pH = 7.44; AA 1000 µM: pH = 7.45 in Krebs–Henseleit (KH) buffer). Finally, 10 µM AA was used in the subsequent experiments.

Additionally, we screened the effects of food ingredients on OATP4C1-mediated transport using AA. The results showed that the overall interactions, especially the inhibitory effects, of food ingredients were not significantly different (Appendix A). However, the inhibitory effects of some food ingredients, such as daidzein, fisetin, genistein, isorhamnetin, luteolin, morin, and quercetin, were weakened in the presence of AA (Appendix A).

### 2.3. Concentration-Dependent Inhibition by Kaempferol and Curcumin

Next, we evaluated the half-maximum inhibitory concentrations (IC_50_) and absolute inhibitory constant (*K*_i_) values of kaempferol and curcumin that inhibited OATP4C1 strongly (Equation (1)). These compounds inhibited OATP4C1-mediated transport in a concentration-dependent manner (Figure 5A–D). The IC_50_ values of kaempferol (AA(−)), kaempferol (AA(+)), curcumin (AA(−)), and curcumin (AA(+)) for OATP4C1 were 29.3, 33.7, 61.1, and 27.5 µM, respectively. The *K*_i_ values of kaempferol (AA(−)), kaempferol (AA(+)), curcumin (AA(−)), and curcumin (AA(+)) for OATP4C1 were 25.1, 28.8, 52.2, and 23.5 µM, respectively. The individual parameters of curcumin are summarized in Table 1. As shown in Figure 5C,D, an evident shift in the concentration-dependent curve was observed for curcumin.

### 2.4. Kinetic Analysis of OATP4C1-Mediated T_3_ Uptake in The Presence or Absence of Kaempferol and Curcumin

Further, we performed a kinetic analysis to clarify the mechanism of inhibition by curcumin and kaempferol. Alteration in the kinetic curves was observed both in the presence of curcumin and kaempferol (Figure 6). The obtained kinetic parameters (*K*_m_, V_max_, and V_max_/*K*_m_ values) are summarized in Table 2. Kinetic analysis revealed that curcumin and kaempferol addition increased the *K*_m_ values of OATP4C1-mediated T_3_ uptake (control, 6.06; curcumin, 20.8; kaempferol, 17.6 (µM)). Moreover, reductions in V_max_/*K*_m_ values were observed in the presence of those inhibitors (control, 88.8; curcumin, 39.7; kaempferol, 31.8 (µL/mg protein/10 min)).

## 3. Discussion

In this study, we aimed to understand the effect of food ingredients on the renal excretion of drugs by investigating their interactions via OATP4C1, which is expressed in renal proximal tubular cells. First, we examined whether 28 types of food ingredients, for which transport inhibition was previously analyzed, inhibited OATP4C1-mediated transport. Twelve compounds, including apigenin, daidzein, fisetin, genistein, isorhamnetin, kaempferol, luteolin, morin, quercetin, curcumin, resveratrol, and ellagic acid, were found to significantly alter OATP4C1-mediated transport. Apigenin, kaempferol, luteolin, morin, quercetin, and curcumin inhibited OATP4C1 transport by ≥50% (Figure 3). Six other compounds also moderately altered OATP4C1-mediated transport: daidzein, fisetin, genistein, isorhamnetin, resveratrol, and ellagic acid (Figure 3). Notably, ellagic acid accelerated OATP4C1-mediated transport by 13% instead of inhibiting it. To date, the structure of OATP4C1 has not been clarified; therefore, the binding sites and intermolecular interactions are unknown. Since the possible existence of multiple binding sites has been reported [18], the differences in the binding sites of each compound might affect the strength of the interactions. Although apigenin, daidzein, genistein, quercetin, gallic acid, resveratrol, and ellagic acid have been reported to inhibit other drug transporters, such as hepatic and intestinal OATPs, renal OATs, and P-glycoprotein (P-gp) (Appendix A), these compounds showed relatively moderate inhibition of OATP4C1 (Figure 3). Theaflavin, epigallocatechin-3-gallate, caffeic acid, ferulic acid, gallic acid, and ellagic acid, which have been reported to inhibit OATP1B1, OATP1B3, OATP2B1, and renal OATs, did not inhibit OATP4C1-mediated transport (Appendix A and Figure 3). Thus, the degree of transport inhibition by food ingredients differed between OATP4C1 and other drug transporters, such as OATPs; however, their details are yet to be elucidated. Moreover, we used much higher concentrations than those expected in human blood to avoid overlooking the interaction. However, the concentrations of the compounds may not reach those used in the inhibition study. Thus, the individual compound may not inhibit OATP4C1 transport by itself in clinical settings.

Next, we performed the transport study using AA to prevent curcumin degradation in the KH buffer. Curcumin can be stabilized by adding antioxidants, such as AA [32]. However, no reports have been made on the use of antioxidants to examine food–drug interactions via transporters. Therefore, we examined the effect of AA on OATP4C1-mediated transport. We found that AA did not affect OATP4C1 transport (Figure 4A). With a reproducibility test, we then confirmed that a significant increase in curcumin inhibition from 33% to 63–67% was observed by preventing curcumin degradation without altering the KH buffer pH (Figure 4B). Notably, curcumin inhibition is not consistent between Figure 3A (72%) and Figure 3B (33%), which may be due to the instability of curcumin in the absence of antioxidants. Furthermore, despite increasing the AA concentration to 100 and 1000 µM, the inhibitory effect of curcumin on transport did not increase compared to the 10-µM AA reaction solution. Thus, 10 µM AA was used in the subsequent experiments. Additionally, we screened the effects of food ingredients on OATP4C1-mediated transport using AA. The results showed that the inhibitory effects of some food ingredients, such as daidzein, fisetin, genistein, isorhamnetin, luteolin, morin, and quercetin, were weakened in the presence of AA (Appendix A). The oxidized metabolites of these compounds may have exerted inhibitory effects on OATP4C1-mediated transport, and the inhibitory effects were weakened as a result of stopping the generation of these metabolites in the presence of AA. Further studies are required to elucidate the mechanisms underlying this phenomenon.

Then, we performed a concentration-dependent study using 10 µM AA. The results showed that kaempferol and curcumin inhibited OATP4C1-mediated transport in a concentration-dependent manner both in the absence and presence of 10 µM AA (Figure 5A–D). The individual parameters of curcumin are summarized in Table 1. An obvious shift was observed in the concentration-dependent curve for curcumin (Figure 5C,D). 

Curcumin degradation is inhibited by binding to plasma proteins, such as albumin, and over 80% of curcumin reportedly remains in the presence of plasma proteins for up to 1 h [33,34]. Curcumin rapidly degrades in phosphate buffer at pH 7.4, with 80–90% decomposition occurring within 12 min, mainly generating bicyclopentadione derivatives through auto-oxidative transformation [32]. Curcumin has also been reported to inhibit OATP1B1 and OATP1B3 transport, with IC_50_ values of 3.8 and 34 µM, respectively [35]. That study did not use antioxidants such as AA, suggesting that oxidative degradation of curcumin might have occurred in the experiments. Based on previous evidence, we successfully evaluated curcumin inhibition of OATP4C1-mediated transport and accurately calculated the IC_50_ and *K*_i_ values using AA. Here, we concluded that the utilization of an antioxidant that does not affect the transporter activity for food–drug interaction studies is desirable to accurately calculate parameters such as IC_50_ and *K*_i_ values.

Finally, we performed a kinetic analysis to clarify the mechanism of inhibition by curcumin and kaempferol. The *K*_m_ value of 6.06 µM in the control was similar to those reported in [18,28]. Alterations in the kinetic curves (Figure 6), increases in the *K*_m_ values, and reductions in the V_max_/*K*_m_ values (Table 2) of OATP4C1-mediated T_3_ uptake were observed in the presence of those inhibitors. The result suggested that these compounds affected OATP4C1-mediated T_3_ uptake in a competitive manner. Moreover, we confirmed that those inhibitors reduced OATP4C1-mediated clearance of T_3_ mainly due to the alteration of *K*_m_ values.

Currently, there is limited information regarding the blood concentrations of food ingredients such as flavonoids. The plasma concentration of flavonoids may not exceed 1 µM with a single flavonoid intake within the range of 10–100 mg [36]. In a normal diet in several countries, the intake of a single flavonoid is reported to be < 100 mg [37,38,39]. Therefore, the blood concentrations of the six food ingredients: apigenin, kaempferol, luteolin, morin, quercetin, and curcumin, identified in this study as OATP4C1 inhibitors, may not be sufficient to inhibit OATP4C1 during normal dietary intake. However, a combination of multiple flavonoids that inhibit P-glycoprotein strengthens its inhibitory effects on the transporter [40]. Therefore, it is possible that additive and/or synergistic inhibitory effects could occur on OATP4C1 when patients consume multiple food ingredients simultaneously. In particular, it is advisable to avoid concurrent intake of kaempferol, curcumin, and substrate drugs of OATP4C1. Furthermore, researchers are developing novel supplements to improve the bioavailability of some flavonoids. Their reports indicated that the blood concentration may exceed 1 µM even when taking a single flavonoid within the range of 10–100 mg [41,42]. When taking such novel supplements, the food–drug interactions via OATP4C1 may occur. However, because information on the pharmacokinetic parameters of food ingredients, such as elimination half-life, is limited, it is challenging to propose specific dosing intervals.

## 4. Materials and Methods

### 4.1. Materials

Ferulic acid (99%) was purchased from Sigma Aldrich (St. Louis, MO, USA). Ascorbic acid and quercetin (≤100%) were purchased from FUJIFILM Wako Pure Chemical Corporation (Osaka, Japan). Caffeic acid (≥98%), curcumin (100%), hesperidin (90%), rutin (95.0–101.5%), and triiodothyronine (T_3_) were purchased from Nacalai Tesque, Inc. (Kyoto, Japan). Gallic acid (≥98%), chlorogenic acid (≥95%), ellagic acid (≥95%), sesamin (≥95%), epigallocatechin-3-gallate (≥96%), genistein (≥98%), daidzein (≥95%), luteolin (≥98%), theaflavin (≥98%), isoquercetin (98%), and morin (100%) were purchased from Cayman Chemical Company (Ann Arbor, MI, USA). Resveratrol (>99.0%), fisetin (>96.0%), isorhamnetin (>95.0%), and troxerutin (90%) were purchased from Tokyo Chemical Industry Co., Ltd. (Tokyo, Japan). Cyanidin chloride (99%) was purchased from Nagara Science Co. Ltd. (Gifu, Japan). Apigenin (≥98%) was purchased from LKT Laboratories, Inc. (St. Paul, MN, USA). Kaempferol (98%), myricetin (98%), and naringin (98%) were purchased from BDL Pharmatech Ltd. (Shanghai, China). Hyperoside (90–100%) was purchased from Toronto Research Chemicals Inc. (Toronto, ON, Canada). Neohesperidin (≥98%) was purchased from Chem-Impex International, Inc. (Wood Dale, IL, USA). All other chemicals were commercially available and had the highest possible purity.

### 4.2. Cell Culture

The MDCKII cells transfected with OATP4C1 or an empty vector were established in our laboratory [18]. OATP4C1/MDCKII and mock cells were cultured in Dulbecco’s modified Eagle’s medium supplemented with 10% fetal bovine serum (Gibco, Thermo Fisher Scientific Inc., Waltham, MA, USA) and G418 (0.5 mg/mL; Nacalai Tesque, Inc.) at 37 °C under 5% CO_2_ and 95% humidified air.

### 4.3. Transport Study

The cellular uptake of T_3_ was measured in monolayer cell cultures grown in 24-well plates. Cells were seeded at a density of 2.0 × 10^5^ cells/well. Thereafter, the cells were incubated in a culture medium containing 5 mM sodium butyrate for 24 h before the uptake study. The cells were washed once with KH buffer (118 mM NaCl, 23.8 mM NaHCO_3_, 4.83 mM KCl, 0.96 mM KH_2_PO_4_, 1.20 mM MgSO_4_, 12.5 mM *N*-(2-hydroxyethyl) piperazine-N’-2-ethanesulfonic acid, 5.0 mM D-glucose, and 1.53 mM CaCl_2_; pH 7.4), followed by preincubation in KH buffer. Cellular uptake was initiated by adding T_3_-containing KH buffer with or without each flavonoid. Uptake was terminated after 10 min by replacing the incubation buffer with ice-cold KH buffer. The cells were washed twice with ice-cold KH buffer. The concentration of T_3_ was measured using liquid chromatography/tandem mass spectrometry (LC/MS/MS). Cellular uptake was presented as the uptake amount divided by the amount of cellular protein quantified using the Bradford protein assay. The experiment was performed in triplicate (n = 3) and repeated three times. Flavonoids, except ellagic acid, were dissolved in dimethyl sulfoxide (DMSO) at a final concentration of less than 0.5%. A stock solution of ellagic acid (20 mM) was obtained by dissolving it in 0.05 M NaOH and was further diluted to 100 µM in KH buffer (the pH was adjusted to 7.4).

### 4.4. Using an Antioxidant to Evaluate the Inhibitory Effects of Food Ingredients

Curcumin is unstable in phosphate buffers at physiological pH [32]. Therefore, AA was used to prevent curcumin degradation during the transport study. Appropriate AA concentrations were determined as follows:(1)The effect of 10, 100, and 1000 µM AA on OATP4C1-mediated transport was evaluated.(2)The inhibitory effect of curcumin was evaluated in the absence or presence of 10, 100, and 1000 µM AA.

Finally, 10 µM AA was determined to be the appropriate concentration of AA and was used in the subsequent transport studies.

### 4.5. Concentration-Dependent Inhibition by Kaempferol and Curcumin

We investigated the concentration-dependent inhibition by kaempferol and curcumin, which resulted in more than 70% inhibition of OATP4C1-mediated T_3_ uptake. The inhibition curve was fitted to the Rodbard model. The half-maximum inhibitory concentration (IC_50_) values of the food ingredients were calculated using JMP Pro 16 (SAS Institute Inc., Cary, NC, USA). The absolute inhibitory constant (*K*_i_) was then derived using the following equation (Equation (1)) [21,43]: where [S] is the substrate concentration and *K*_m_ is the Michaelis–Menten constant. The experiment was performed in triplicate (n = 3) and repeated three times.
(1)$$K_{i}=\frac{{IC}_{50}}{1+\frac{\left[S\right]}{K_{m}}}$$

### 4.6. Kinetic Analysis of OATP4C1-Mediated T_3_ Uptake in the Presence or Absence of Kaempferol and Curcumin

We performed the kinetic analysis to clarify the mechanism of inhibition by curcumin and kaempferol, which showed strong inhibition on OATP4C1-mediated T_3_ uptake. The kinetic curves were fitted to the Michaelis–Menten equation. The kinetic parameters (*K*_m_, V_max_, and V_max_/*K*_m_ values) of the food ingredients were calculated using JMP Pro 16. The experiment was performed in duplicate (n = 2) and repeated three times.

### 4.7. Sample Preparation

The cells were scraped and homogenized in 200 µL water after uptake termination. Deproteinization was performed by adding equal volumes of pravastatin-containing acetonitrile as an internal standard. After vortexing, the mixture was centrifuged at 15,000× *g* for 5 min at 20 °C, and the supernatant was collected.

### 4.8. Liquid Chromatography/Tandem Mass Spectrometry Conditions

LC/MS/MS was used to measure the T_3_ concentrations. A Shimadzu Nexera HPLC System (Shimadzu Corporation, Kyoto, Japan) equipped with a Cosmosil 5C18-MS-II column (Nacalai Tesque, Inc.) was used for chromatographic separation. The mobile phase was water/acetonitrile (70:30, *v/v*) containing 0.1% acetic acid at a flow rate of 0.2 mL/min. The column temperature was maintained at 40 °C and the sample injection volume was 5 µL.

Mass spectrometry was performed using an API 5000 tandem mass spectrometer (Sciex LLC, Framingham, MA, USA). The T_3_ concentration was measured in negative ion mode. Selected reaction monitoring (SRM) was performed and the SRM transitions monitored were *m/z* 650 > 127 for T_3_ and *m/z* 423 > 101 for pravastatin. The collected data were analyzed using the Analyst software (version 1.5; SCIEX LLC).

### 4.9. Statistical Analysis

Data are expressed as the mean ± SE. Multiple statistical comparisons were made using one-way analysis of variance (ANOVA), followed by Tukey’s test. The data were analyzed using JMP Pro 16 (SAS Institute Inc.). Statistical significance was set at *p*-values < 0.05.

## 5. Conclusions

We clarified food–drug interactions via the renal drug transporter OATP4C1 and discussed the differences between other drug transporters and OATP4C1. This is the first report to propose the utilization of an antioxidant for the evaluation of food–drug interactions via transporters. Our findings demonstrated that the addition of ascorbic acid was beneficial in evaluating the effect of curcumin on OATP4C1-mediated transport. Our results suggest the usefulness of an antioxidant such as ascorbic acid when assessing the interaction of easily oxidized compounds on transporters or metabolic enzymes. We believe that this study provides novel information for food–drug interaction studies of drug transporters.

## Figures and Tables

**Figure 1 ijms-25-00991-f001:**
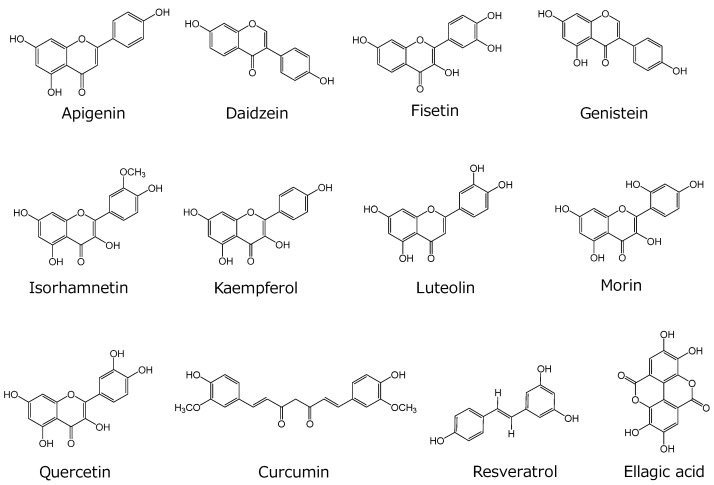
Chemical structures of the tested food ingredients that have strong and moderate alterations in OATP4C1-mediated transport.

**Figure 2 ijms-25-00991-f002:**
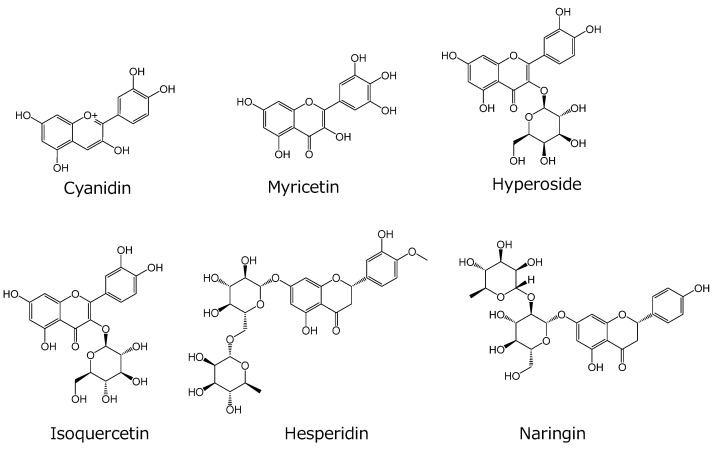
Chemical structures of the tested food ingredients that did not exhibit significant alterations in OATP4C1-mediated transport.

**Figure 3 ijms-25-00991-f003:**
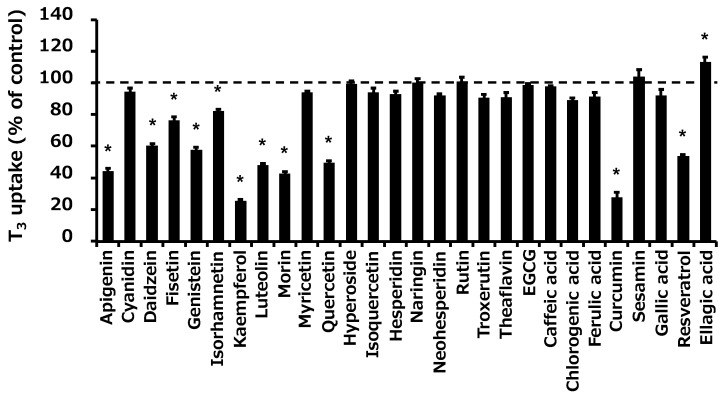
Screening of the effect of food ingredients on OATP4C1-mediated transport. Cells were incubated with 1 µM T_3_ with or without food ingredients for 10 min at 37 °C. The concentration of the compounds was 100 µM, except for theaflavin (50 µM). The amount of OATP4C1-mediated uptake was calculated by subtracting the nonspecific uptake of T_3_ by mock cells from the total cellular uptake by OATP4C1-expressing cells. Data are presented as the mean ± standard error (SE) (n = 3). * *p* < 0.05, significantly different from the control by one-way analysis of variance followed by Tukey’s test. Dashed line represents 100% (control uptake in the absence of food ingredient).

**Figure 4 ijms-25-00991-f004:**
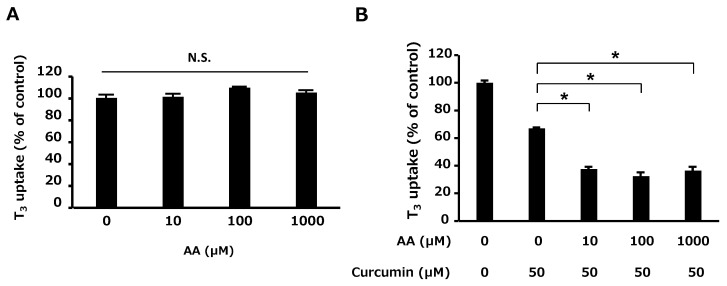
Inhibitory effects of curcumin on OATP4C1-mediated transport under the presence of AA. (**A**) Cells were incubated with 1 µM T_3_ with AA for 10 min at 37 °C. The AA concentrations were 0, 10, 100, and 1000 µM. (**B**) Cells were incubated with 1 µM T_3_ and 50 µM curcumin with or without 10, 100, and 1000 µM AA for 10 min at 37 °C. The amount of OATP4C1-mediated uptake was calculated by subtracting the nonspecific uptake of T_3_ by mock cells from the total cellular uptake by the OATP4C1-expressing cells. Data are shown as the mean ± SE (n = 3). N.S., not significantly different. * *p* < 0.05, significantly different.

**Figure 5 ijms-25-00991-f005:**
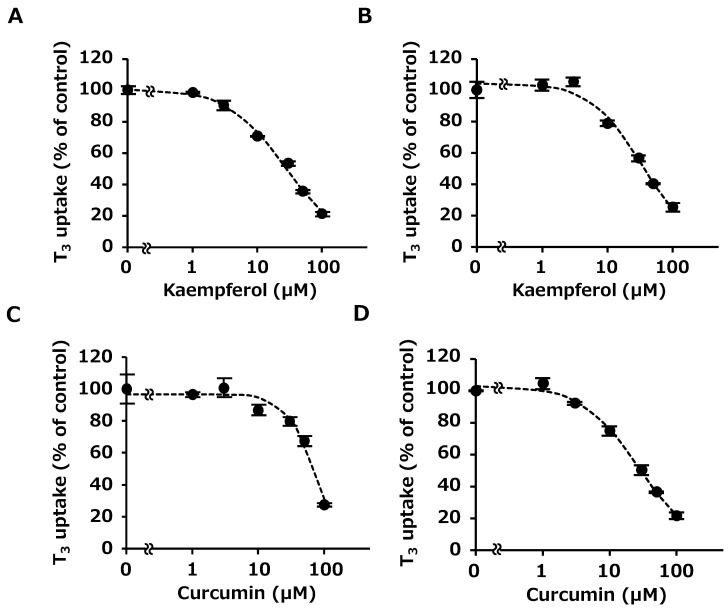
Concentration-dependent inhibition of OATP4C1-mediated T_3_ transport by kaempferol (AA(−)) (**A**), kaempferol (AA(+)) (**B**), curcumin (AA(−)) (**C**), and curcumin (AA(+)) (**D**). Cells were incubated with 1 µM T_3_ with kaempferol or curcumin with or without 10 µM AA for 10 min at 37 °C. Their concentrations were 0, 1, 3, 10, 30, 50, and 100 µM. The amount of OATP4C1-mediated uptake was calculated by subtracting the nonspecific uptake of T_3_ by mock cells from the total cellular uptake by OATP4C1-expressing cells. Data are shown as the mean ± SE (n = 3). Dots and bars represent the mean ± SE (n = 3). The dashed lines are the inhibition curves fitted by using the Rodbard model. Double tilde indicates omission between 0 and 1.

**Figure 6 ijms-25-00991-f006:**
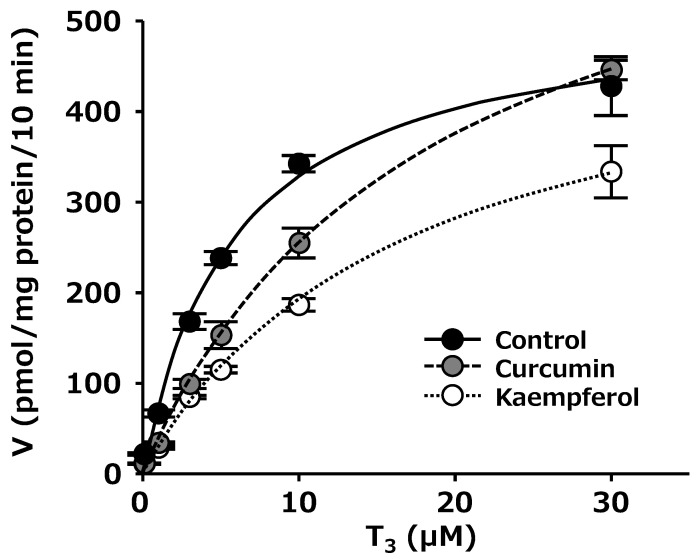
Concentration dependence of OATP4C1-mediated T_3_ uptake in the presence or absence of kaempferol and curcumin. Cells were incubated with 0.3, 1, 3, 5, 10, and 30 µM T_3_ with or without the inhibitors (kaempferol (30 µM) and curcumin (30 µM)) in the presence of 10 µM AA for 10 min at 37 °C. The amount of OATP4C1-mediated uptake was calculated by subtracting the nonspecific uptake of T_3_ by mock cells from the total cellular uptake by OATP4C1-expressing cells. Data are shown as the mean ± SE (n = 6). The kinetic curves were fitted to the Michaelis–Menten equation. The black, gray, and white circles were the control, curcumin, and kaempferol, respectively. Each curve represents the fitted line obtained by the kinetic analysis by JMP Pro 16.

**Table 1 ijms-25-00991-t001:** IC_50_ and *K*_i_ values of curcumin in the absence or presence of AA.

	IC_50_ (µM)	*K*_i_ (µM)	Average of *K*_i_ (µM)	SE (µM)
AA (−)	49.4	42.2	52.2	5.1
	69.4	59.3		
	64.5	55.2		
AA (+)	26.5	22.7	23.5	0.5
	28.6	24.5		
	27.4	23.4		

**Table 2 ijms-25-00991-t002:** *K*_m_, V_max_, and V_max_/*K*_m_ values of OATP4C1-mediated T_3_ uptake in the absence or presence of kaempferol and curcumin.

	*K*m (µM)	Vmax (pmol/mg Protein/10 min)	Vmax/*K*m (µL/mg Protein/10 min)
Control	6.06 ± 0.83	524 ± 18.2	88.8 ± 8.90
Curcumin	20.8 ± 4.89	757 ± 76.0	39.7 ± 6.79
Kaempferol	17.6 ± 5.01	517 ± 92.6	31.8 ± 4.16

## Data Availability

The authors declare that all the data supporting the findings of this study are available within the paper and its Appendix A Data.

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
