# Peer review of "The Use of an Antioxidant Enables Accurate Evaluation of the Interaction of Curcumin on Organic Anion-Transporting Polypeptides 4C1 by Preventing Auto-Oxidation"

_ijms, 2024, doi:10.3390/ijms25020991_

Round 1

Reviewer 1 Report

Comments and Suggestions for Authors

In the article entitled "Screening of food-drug interactions via organic anion-trans-porting polypeptides 4C1 in the presence of an antioxidant”. The authors highlight the importance of the use of the antioxidant in food-drug interaction via OATP4C1. The manuscript is well written, and the outcomes are presented clearly. However, there are some suggestions to further improve the manuscript:

1-     I suggest the authors improve the title of the paper since it doesn´t summarize the main idea of the study. It should be more accurate and interesting.

2-     Please define IC50 abbreviation for the first time used.

3-     I suggest dividing Figure 1 into two Figures. Figure 1 refers to the chemical structures of tested food ingredients that have a strong and moderate inhibition. And Figure 2 refers to other chemical structures of tested food ingredients that have a low inhibition.

4-     The legend of Figure 2 should include the p-value. And why is there asterisk above the bar of Ellagic acid?

5-     On page 7, from lines 164 to 166, the phrase “Moreover, because the concentrations… remains.” The phrase is unclear. Please rephrase this phrase so that it can be understood better by the reader.

6-     On page 8, from lines 212 to 214, the phrase “Therefore, it is possible…inhibit OATP4C1.” The phrase is unclear. Please rephrase this phrase so that it can be understood better by the reader.

7-     On page 8, from lines 225 to 227, the phrase “Our findings demonstrated…such a curcumin.” The phrase is unclear. Please rephrase this phrase so that it can be understood better by the reader.

8-     The paragraph of conclusion should be removed from the discussion section. And the authors should add apart the section of conclusion to briefly summarize the outcomes of this work.

9-     There are some references that need to be updated.

10-  p. 1, line 32: The term "under" suggests that there must be "in".

11-  p. 1, line 34: The term "shall" suggest that there must be "will".

12-  p. 2, lines 70-71: The term "substrate or inhibitor" suggest that there must be "substrates or inhibitors".

13-  p. 7, line 180: The term "utilizing" suggest that there must be "using".

14-  p. 8, line 214: The term "intake" suggests that there must be "intakes".

15-  p. 9, lines 265-287: The term "thrice" suggest that there must be "three times".

16-  p. 9, line 267: The term "Stock" suggests that there must be "A stock".

17-  p. 9, lines 266-267: The term "of DMSO was less than" suggest that there must be  "less than".

Comments on the Quality of English Language

In the manuscript entitled "Screening of food-drug interactions via organic anion-trans-porting polypeptides 4C1 in the presence of an antioxidant". The authors have written the article in comprehensive english with minimal grammatical errors.

Reviewer 2 Report

Comments and Suggestions for Authors

Review report attached

Round 2

Reviewer 2 Report

Comments and Suggestions for Authors

I am fine with the Authors' revision. 

Author Response

It was my pleasure that I fulfilled your requests to further improve my manuscript. Thank you again for reviewing.